# Diatom Biodiversity in Karst Springs of Mediterranean Geographic Areas with Contrasting Characteristics: Islands vs Mainland

**Giuseppina G. Lai** [1,*][iD]**, Sara Burato** [2][iD]**, Bachisio M. Padedda** [1,*][iD]**, Raffaella Zorza** [2][iD]**, Elisabetta Pizzul** [3][iD]**, Cristina Delgado** [4][iD]**, Antonella Lugliè** [1][iD] **and Marco Cantonati** [5][iD]

1   Department of Architecture, Design and Urban Planning, University of Sassari, via Piandanna 4, 07100 Sassari, Italy; luglie@uniss.it
2   ARPA FVG (Regional Agency for Environmental Protection Friuli Venezia-Giulia), Via Cairoli 14, 33057 Palmanova, Italy; sara.burato@alice.it (S.B.); raffaella.zorza@arpa.fvg.it (R.Z.)
3   Department of Life Sciences, University of Trieste, via Giorgieri 10, 34127 Trieste, Italy; pizzul@units.it
4   Department of Ecology and Animal Biology, University of Vigo, Campus As Lagoas Marcosende, 36330 Vigo, Spain; cdelgado.cristina@gmail.com
5   MUSE–Museo delle Scienze, Limnology & Phycology Section, Corso del Lavoro e della Scienza 3, 38122 Trento, Italy; marco.cantonati@muse.it
*   Correspondence: laigg@uniss.it (G.G.L.); bmpadedda@uniss.it (B.M.P.); Tel.: +39-079-213042 (G.G.L.); +39-079-228670 (B.M.P.)

**Abstract:** Karst ecosystems are considered as priority environments for the protection of biodiversity on a global scale. This study provides a first comparative analysis of epilithic diatom flora from karst springs in two Mediterranean geographic areas (Spain and Italy) with contrasting characteristics (islands vs mainland). We investigated twenty-three springs with different anthropogenic impact levels once in the winter season between 2007 and 2017 (N = 23). A total of 176 diatom taxa (56 genera) were found of which 101 (44 genera) were observed in single sites. A general good biotic integrity was revealed by structural indices (species richness, diversity and evenness). However, crenophilous species were generally present and abundant in less impacted springs. Comparing islands and mainland, significant differences were found in species composition and diversity (H′) based on multivariate analyses (global R = 0.610; $p$ = 0.001) and $t$-test ($t$ = 2.304; $p$ = 0.031). Discharge and Cl⁻ were the most significant variables in determining diatom assemblages. Our results confirm the role of springs as multiple ecotones and refuges for rare species and suggest that the geographic insularity may be an important factor in maintaining diatom biodiversity.

**Keywords:** Bacillariophyceae; microalgae; biological diversity; indicator species; carbonate substrate; karst waters

## 1. Introduction

Karst springs are freshwater environments of great ecological value and strategic water resources in the Mediterranean area since they provide valuable ecosystem services, among which habitat for high biodiversity and drinking water supply [1]. In this geographic area, due to the strong climatic seasonality and lower water availability during dry periods (summer–autumn), several large springs have been captured since ancient times and they are almost the exclusive drinking water source for many urban centres [2,3].

Karst ecosystems have been indicated as priority environments for the protection of biodiversity on a global scale [4]. Unlike other freshwater ecosystems, several springs are still relatively natural

environments [5] and can provide specific habitats for crenobiontic and crenophilous species [6,7]. As multiple ecotones, they include mosaic-like structures or patches of different aquatic, semi-aquatic and semi-terrestrial microhabitats (e.g., mosses and debris layers) which can host a large number of species and serve as refuges for rare and threatened organisms and least-impaired habitat relicts (LIHRe) [7,8].

Karst springs are intrinsically sensitive to variations in the hydrological cycle, water abstraction and pollution [9,10], and are more exposed to these impacts in the Mediterranean area because of the variable rainfall regime, climate change and increasing water use e.g., [11,12]. Hydromorphological alterations in these ecosystems can make some species vulnerable, especially those with limited dispersal, and cause an impoverishment of the biota, in particular, aquatic organisms. In fact, the quality and quantity of water and substrates are the main factors influencing the development of aquatic plant communities [13]. Further, habitat alteration is one of the major drivers of species loss [7]. Biodiversity, especially of microorganisms, play an important role in abiotic processes and maintenance of karst systems [14]. Recent studies highlighted the need for integrated multidisciplinary approaches, also based on biology and ecology, for sustainable use and proper management of karst springs [1,15].

Diatoms are an abundant group of microalgae in springs [16]. As primary producers they sustain the food webs in these ecosystems, significantly contributing to their functioning [17,18]. They are also useful indicators of important environmental features of springs, such as water quality (chemical composition and trophic conditions), water flow permanence, hydraulic regime, discharge, shading and lithology of the aquifers e.g., [19,20].

In the Mediterranean area, studies on diatoms from karst springs were carried out mainly in mountain regions in Slovenia [21], Croatia [22], Bosnia and Herzegovina [23], and Italy [18,24,25]. Only a few studies focused on low-altitude springs on islands: Majorca, the largest of the Balearic Islands (Spain) [26] and Sardinia (Italy), the second-largest island in the Mediterranean basin [27–32]. In general, these studies pointed out high levels of biodiversity and the presence of species new to science (e.g., *Navicula veronesis* Lange-Bertalot & Cantonati and *Sellaphora gologonica* G.G. Lai, Ector & C.E. Wetzel). They also provided information on the ecology of species and their relationships with environmental variables, considering both natural and anthropogenic disturbances, and suggested good prospects in the use of diatoms for the definition of ecological integrity and vulnerability aspects of karst springs. However, these studies were limited and focused on a single spring or groups of springs in specific areas, and comparisons on a wider geographic scale are not available in the literature.

This study compares the epilithic diatom flora from karst springs of four regions in two Mediterranean geographic areas (Spain and Italy). We hypothesized significant differences among diatom assemblages due to contrasting characteristics (islands vs mainland) and local factors. The main objectives were: (1) to examine the species composition and structure of diatom assemblages; (2) to evaluate similarities/dissimilarities among geographic areas; (3) to explore relationships between diatom taxa and environmental variables.

## 2. Materials and Methods

### 2.1. Study Sites

For this study, 23 springs were selected from four datasets built on a regional scale: Majorca Island (Spain), and Sardinia, Veneto and Friuli (Italy) (Figure 1, Table 1).

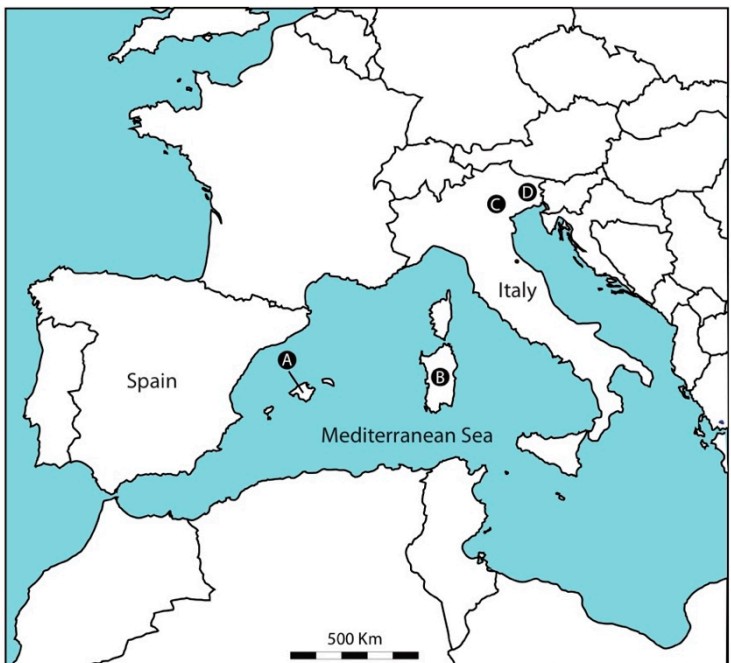

**Figure 1.** Geographic position of the four regions studied in the two Mediterranean areas: (A) Majorca Island (Spain); (B) Sardinia, (C) Veneto, (D) Friuli (Italy).

**Table 1.** Main characteristics of the springs studied. Type: R = Rheocrenic; R-L = Rheo-limnocrenic; L-R = Limno- rheocrenic. Regime: P = Perennial; S = Seasonal.

| Spring Name | Code | Altitude (m.a.s.l.) | Latitude (N) | Longitude (E) | Type | Regime | Spring-Tapping |
|---|---|---|---|---|---|---|---|
| **Majorca** | | | | | | | |
| Font des Pí | FP | 471 | 39°46′14″ | 2°48′26″ | R | P | X |
| Font de s' Olla | FL | 56 | 39°45′23″ | 2°42′42″ | R | P | X |
| Son Vich | SV | 356 | 39°39′12″ | 2°32′08″ | R | P | X |
| **Sardinia** | | | | | | | |
| Fruncu 'e Oche | FO | 55 | 40°33′84″ | 009°40′67″ | R | P | X |
| Pubusinu | PB | 212 | 39°24′34″ | 008°31′39″ | R | P | X |
| San Giovanni-DN | DN | 190 | 39°20′13″ | 008°37′37″ | R | P | X |
| San Giovanni-DR | DR | 168 | 40°19′14″ | 009°36′55″ | R | P | X |
| Sa Varva | VA | 780 | 40°13′39″ | 009°12′31″ | R | S | |
| Sa Vena Manna-S | MS | 276 | 40°50′25″ | 008°48′20″ | R | P | X |
| Su Gologone-SVM | SG | 116 | 40°17′34″ | 009°29′80″ | L-R | P | |
| S'Ulidone | UL | 606 | 40°34′30″ | 009°38′35″ | R | P | X |
| **Veneto** | | | | | | | |
| Brusaferri | BR | 675 | 45°35′57″ | 11°12′59″ | R-L | P | X |
| Buso delle Anguane | BA | 370 | 45°33′04″ | 11°17′07″ | R-L | P | X |
| Fonte del Coppo | FC | 183 | 45°28′39″ | 11°06′51″ | L-R | P | X |
| La Ferrara | LF | 640 | 45°32′59″ | 11°11′08″ | L-R | P | X |
| Prealba | PR | 445 | 45°37′34″ | 10°49′47″ | L-R | P | X |
| Torricelle | TO | 220 | 45°28′09″ | 11°00′16″ | R-L | P | X |
| **Friuli** | | | | | | | |
| Torre | TR | 528 | 46°18′27″ | 13°16′21″ | R | P | |
| Uccea | UC | 1.122 | 46°20′02″ | 13°19′30″ | R | P | |
| Barman | BM | 652 | 46°20′21″ | 13°17′18″ | R | P | |
| Rio Researtico | RR | 454 | 46°22′22″ | 13°13′01″ | R | P | X |
| Goriuda | GD | 701 | 46°23′44″ | 13°26′8″ | R | P | |
| Alba | AL | 1.249 | 46°28′12″ | 13°13′48″ | R | P | |

The three springs studied in Majorca Island are located in the Tramuntana mountain range, in the northern part of the island, and cover an altitudinal range of 56–471 m a.s.l. The Tramuntana Mountains extend parallel to the north-western coast and present numerous springs with different

sizes and discharge [33]. The springs studied in this area are perennial and rheocrenic systems, and some of them are captured for drinking purposes.

In Sardinia, the eight springs studied are located in different areas of the island and distributed over an elevational range of 55–780 m a.s.l. They are mostly perennial and rheocrenic systems. Only two springs, SG and VA are classified as limno-rheocrenic and seasonal, respectively. The springs SG and DN are "Natural Monuments" according to Regional Law 31/1998 that defines the areas of noteworthy naturalistic value for conservation and protection in Sardinia. Five springs are located in Sites of Community Importance (SCI) for the Mediterranean biogeographic region: UL and FO (ITB 021107 "Monte Albo") SG (ITB022212 "Supramonte di Oliena, Orgosolo e Urzulei–Su Sercone"), VA (ITB021156 "Monte Gonare") and DN (ITB041111 "Monte Linas–Marganai"). Most of these springs are captured for drinking purposes.

In the Veneto Region, almost all springs (five out of six) are located in the south-eastern Alpine foothills, in a geographic area called Lessinia. One spring (PR) is located on the Monte Baldo mountain range (south-eastern Alps). They cover an elevation range of 183–675 m a.s.l. These springs are perennial systems and are virtually all captured (tapped) to some extent. Three springs (BR, BA, and TO) are classified as rheo-limnocrenic, and three springs (FC, LF and PR) as limno-rheocrenic systems. This is typically due to artificial (concrete) water collection basins that occupy part of the spring-head.

Finally, in Friuli, the six springs studied cover an altitudinal range of 454–1249 m a.s.l and are perennial and typical rheocrenic systems. The springs BM and GD are included in the list of geosites of regional and national interest, respectively [34]. The site GD is classified as "Alpine Cave" (habitat code SC1) [35]. The springs RR is captured for drinking purposes.

## 2.2. Sampling

Sampling was carried out once in the winter season between 2007 and 2017 (N = 23). The geographic position and altitude of the springs were recorded by a GPS. Water temperature, pH and conductivity were measured in situ using digital multi-parametric probes [24,26,32].

Anthropogenic disturbances were evaluated in an ordinal scale with disturbance scores: 1 = low; 2 = medium; 3 = high [24]. Shading conditions and current velocity were assessed using a five-score scale [36]. In most springs, discharge was measured by the volumetric method using a graduated bucket and a chronometer. Repeated measures were made close to the diatom sampling points. Only in some large springs (TR, BM and AL), discharge was obtained by estimates based on current velocity/area ratio. The discharge measured during sampling was reported using a seven-range scale (<0.5; 0.5–1; 1–5; 5–10; 10–50; >50; >100 L sec$^{-1}$).

Diatom and water samples for physical and chemical analyses were collected simultaneously. Diatoms were collected with a toothbrush from hard natural substrata (five stones) along a longitudinal transect from the emergence point up to a max distance of 5 m following the European Standard [37]. All samples were preserved with a formaldehyde solution (4% v/v) immediately after sampling. Some typically selected sampling points in each geographic region are reported in Figure 2.

Water samples were collected at the emergence point using 1-L polypropylene and polyethylene bottles and were transported in cold and dark conditions for the laboratory analyses. Further details for the methods used are given in previous studies [24,26,32].

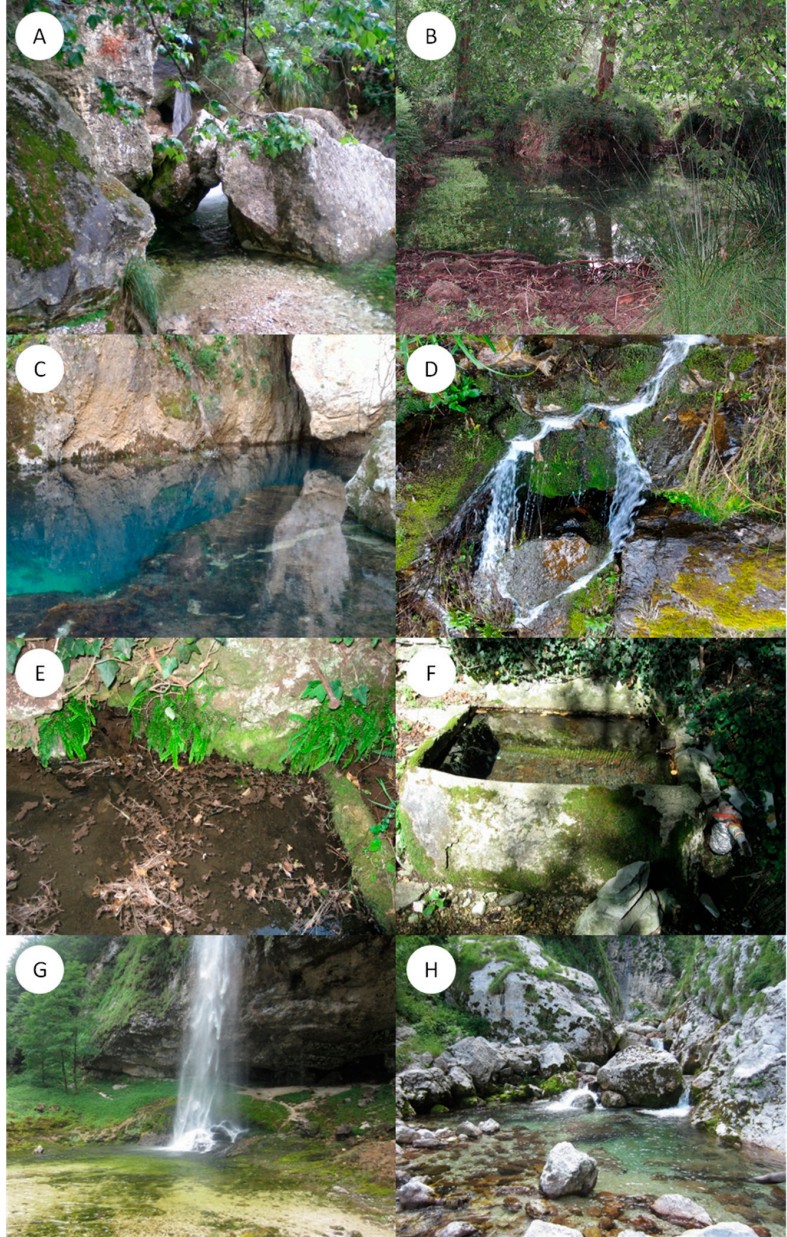

**Figure 2.** Some typically selected sampling points: (**A**) Font des Pí; (**B**) Son Vich; (**C**) Su Gologone-SVM; (**D**) S'Ulidone; (**E**) Buso delle Anguane; (**F**) La Ferrara; (**G**) Goriuda; (**H**) Barman.

*2.3. Laboratory Analyses*

Water samples were analysed in a laboratory using standard methods [38–40]. Diatom samples were treated to obtain a suspension of clean frustules. Organic matter was eliminated by an oxidation process on a heating plate using hydrogen peroxide ($H_2O_2$) and hydrochloric acid (HCl) was added to remove carbonates [41]. After rinsing with distilled water, permanent slides were mounted using Styrax (refractive index = 1.59) and Naphrax (refractive index = 1.73) mounting medium. Diatom observations and counts (~400 valves/frustules for each slide) were performed using a light microscope equipped with phase contrast and micrometric scale at ×1000 magnification. Diatom species were identified according to classic European floristic books [20,42–47], and recent taxonomic literature e.g., [48–51].

*2.4. Data Processing and Statistical Analyses*

The structure of diatom assemblages was examined by species richness (*R*), Shannon-Wiener diversity index (*H'*) [52] and Pielou's evenness index (*J'*) [53], calculated using OMNIDIA 6.0 software [54]. Differences in structural indices between islands and the mainland were evaluated by a paired *t*-test [55,56] and after verification of normal distribution with the Shapiro-Wilktest [57] using R 3.6.1 [58]. For this test probability (*p*) <0.05 was considered significant.

All diatom data were converted in relative abundances (RA) for the statistical analyses. To minimize the influence of rare species, only the species with RA ≥ 5% in at least two sites are included in the analyses.

Ecological preferences of the most abundant diatom species (RA ≥ 5%) for pH, trophic state, salinity and moisture were attributed according to Van Dam et al. [59]. The taxa with a wider ecological range were equally placed among the respective autecological levels to obtain a simplified framework.

The similarity among diatom assemblages of the four geographic regions was analysed using a non-metric multidimensional scaling ordination (nMDS) [60]. The ordination was performed on a Bray-Curtis similarity matrix of species data log (x+1) transformed [61]. The significance of the differences was tested by a one-way analysis of similarities (ANOSIM) [62]. For this analysis, $p < 0.05$ was considered significant. The ANOSIM pairwise test was also performed for each pair of geographic regions. nMDS and ANOSIM were performed using the software PRIMER 6 [63].

Relationships between diatom species and environmental variables were explored by a Canonical Correspondence Analysis (CCA) [64] after the previous assessment of the length of the gradient (>4) by means of a Detrended Correspondence Analysis (DCA) [65] of diatom data using Canoco 4.5 [66]. The two matrices, for physical and chemical data (except for pH), and for diatom data, were log (x + 1) transformed. All canonical axes were used to assess the significant variables through analyses by means of a Monte Carlo test (499 permutations).

The indicator value method (IndVal) [67] was used to identify the most characteristic species of three groups defined by the CCA. Indicator values were calculated using abundance and frequency of occurrence of each species in each group as the input data. The species with $p < 0.05$ based on 499 permutations from the Monte Carlo test were considered to be indicator species. The IndVal was performed by means of the R package indicspecies (ver. 1.7.1).

## 3. Results

*3.1. Environmental Variables*

The values of the environmental variables measured and analysed for all springs studied are presented in Table 2. The water temperature ranged from 6.7 to 16.4 °C. The lowest values (6.7–8.7 °C) were measured at BM, AL, UC and TR in Friuli and the highest values (15.1–16.4 °C) at FO, PB and MS in Sardinia and SV and FL in Majorca. pH was slightly alkaline (>7) at several sites and circumneutral (slightly <7) at FP, SV in Majorca, and at VA in Sardinia. More alkaline values of pH (8.24–8.71) were recorded for springs in Friuli and for only one spring (DN) in Sardinia. Conductivity was low-intermediate in all sites (131–1140 $\mu$S cm$^{-1}$). Impacts values (score > 6) were recorded at several springs, mainly in Veneto and Sardinia. Higher shading values (score ≥ 4) were recorded at five springs: FL and SV in Majorca, VA in Sardinia and FC and BR in Veneto. Discharge and current velocity showed very different values among sites. Higher discharge (up to >100 L s$^{-1}$) and current velocity (score = 4) were generally recorded in the springs of Friuli. The P-PO$_4$$^{3-}$ values, available for all sites except for Veneto, were higher at DR (161 $\mu$g L$^{-1}$) and MS (66 $\mu$g L$^{-1}$) in Sardinia. The N-NO$_3$$^-$ concentrations were higher for springs in Veneto (1580–5869 $\mu$g L$^{-1}$) and Friuli (2300–6200 $\mu$g L$^{-1}$). Cl$^-$ ranged from 0.4 to 195 mg L$^{-1}$ with lower values and narrower range for springs in Friuli (0.4–4.1 mg L$^{-1}$) and higher values and a wider range in Majorca (15.7–45.0 mg L$^{-1}$) and Sardinia (21.3–195 mg L$^{-1}$). Ca$^{2+}$ and Mg$^{2+}$ showed higher values in Veneto (respectively, 52.4–111.5 mg L$^{-1}$ and 13.6–52.7 mg L$^{-1}$) and Majorca (respectively, 75.3–97.5 mg L$^{-1}$ and 13.0–35.1 mg L$^{-1}$).

**Table 2.** Values of the environmental variables measured and analysed in all springs studied.

| Variable/Spring | Majorca | | | Sardinia | | | | | | | | Veneto | | | | | | | Friuli | | | | |
|---|---|---|---|---|---|---|---|---|---|---|---|---|---|---|---|---|---|---|---|---|---|---|---|
| | FL | FP | SV | FO | PB | DN | DR | VA | MS | SG | UL | FC | BA | PR | TO | LF | BR | TR | UC | BM | RR | GD | AL |
| T [°C] | 15.7 | 12.5 | 15.1 | 15.1 | 15.9 | 14.2 | 13.4 | 13.2 | 16.4 | 13.0 | 9.5 | 13 | 12.2 | 11.9 | 12 | 12.3 | 10.3 | 8.7 | 8.3 | 6.7 | 10.8 | 13.1 | 7.3 |
| pH [units] | 7.25 | 6.65 | 6.91 | 7.62 | 7.80 | 8.10 | 7.74 | 6.98 | 7.49 | 7.52 | 7.20 | 7.83 | 7.9 | 7.71 | 7.6 | 7.2 | 7.2 | 8.32 | 8.24 | 8.71 | 8.38 | 8.47 | 8.40 |
| Conductivity [$\mu$S cm$^{-1}$] | 612 | 416 | 796 | 339 | 571 | 478 | 439 | 760 | 1140 | 326 | 357 | 424 | 333 | 438 | 436 | 498 | 578 | 154 | 170 | 136 | 148 | 131 | 229 |
| Impacts [score] | 4 | 1 | 3 | 8 | 7 | 8 | 9 | 3 | 9 | 3 | 3 | 10 | 5 | 10 | 3 | 10 | 5 | 7 | 5 | 2 | 3 | 3 | 2 |
| Shading [score] | 5 | 2 | 4 | 3 | 2 | 3 | 2 | 4 | 1 | 3 | 3 | 5 | 2 | 2 | 2 | 3 | 4 | 1 | 1 | 3 | 1 | 3 | 1 |
| Discharge [L s$^{-1}$] | 10–50 | 0.5–1 | 5–10 | 0.5–1 | <0.5 | <0.5 | <0.5 | 0.5–1 | <0.5 | 1–5 | <0.5 | <0.5 | <0.5 | 0.5–1 | <0.5 | <0.5 | 0.02 | >100 | 10–50 | >100 | 5–10 | 0.5–1 | >50 |
| Current velocity [score] | 4 | 2 | 3 | 3 | 2 | 2 | 2 | 3 | 2 | 3 | 3 | 3 | 2 | 2 | 2 | 3 | 2 | 4 | 4 | 4 | 4 | 1 | 4 |
| P-PO$_4{}^{3-}$ [$\mu$g L$^{-1}$] | 1 | 1 | 1 | 14 | 7 | 8 | 161 | 8 | 66 | 10 | 9 | - | - | - | - | - | - | 0 | 10 | 0 | 10 | 20 | 0 |
| N-NO$_3{}^{-}$ [$\mu$g L$^{-1}$] | 225 | 134 | 231 | 371 | 460 | 309 | 2093 | 1549 | 3634 | 806 | 49 | 2257 | 1354 | 2257 | 5643 | 5869 | 1580 | 3000 | 2300 | 2300 | 2300 | 6200 | 3000 |
| Cl$^{-}$ [mg L$^{-1}$] | 24.1 | 15.7 | 45.0 | 28.4 | 56.7 | 60.3 | 39.0 | 42.5 | 195.0 | 21.3 | 46.0 | 7.8 | 15.0 | 9.9 | 11.7 | 15.0 | 6.0 | 1.0 | 0.5 | 0.4 | 0.6 | 4.1 | 0.5 |
| Ca$^{2+}$ [mg L$^{-1}$] | 97.5 | 75.3 | 85.8 | 30.0 | 42.0 | 32.0 | 48.0 | 56.0 | 54.0 | 32.0 | 18.0 | - | 52.4 | - | - | 106.2 | 111.5 | 24.4 | 35.8 | 23.7 | 21.3 | 25.4 | 22.0 |
| Mg$^{2+}$ [mg L$^{-1}$] | 22.5 | 13.0 | 35.1 | 18.2 | 27.9 | 21.9 | 18.2 | 51.0 | 48.6 | 19.4 | 18.2 | - | 13.6 | - | - | 52.7 | 15.7 | 7.4 | 1.0 | 4.6 | 8.9 | 2.2 | 12.8 |

### 3.2. Species Composition and Structure of Diatom Assemblages

The complete list of species was reported in the supplementary table. A total of 176 taxa from 56 genera were found of which 5 (5 genera) were centric and 171 (51 genera) pennate. *Ellerbeckia arenaria* was the most abundant centric species with a maximum RA = 5.3%. The most rich-species genera were *Gomphonema* (20), *Nitzschia* (17), and *Navicula* (14), followed by *Achnanthidium* (10), *Diploneis* (9), and *Sellaphora* (6). Overall, the species common to samples collected in the islands were 22 and those common to samples collected in the mainland were 11.

The most abundant taxa (RA ≥ 5% in at least one site) were 46 from 25 genera (26% of the total species) (Table 3). Among these, the most frequent species (i.e., occurring in more than six sites) were 13: *Achnanthidium lineare*, *A. minutissimum, A. pyrenaicum, Amphora pediculus, Caloneis fontinalis, Cocconeis euglypta*, *C. pseudolineata, Denticula tenuis, Humidophila contenta, Meridion circulare, Planothidium frequentissimum*, *P. lanceolatum* and *Sellaphora nigri*. Considering the most abundant taxa, ecological information was available for 24 taxa (pH and trophic state), 26 taxa (salinity) and 21 taxa (moisture). These species are mainly alkaliphilous (63%), oligohalobous (65%), and linked to the aquatic environment (86%). The trophic requirements highlighted a prevalence of species characteristic of eutrophic and mesotrophic environments, respectively 46% and 29%, in respect of species characteristic of oligotrophic environments (25%).

The taxa found in single sites were 101 from 44 genera (57% of the total species). Among these, the most abundant were *Cymbella tridentina*, *Eunotia arcubus*, *Fallacia insociabilis*, *F. muraloides*, *Gomphonema* aff. *cymbelliclinum*, *Hannaea arcus*, *Nitzschia frustulum*, *Platessa conspicua* and *Psammothidium grischunum*. The number of taxa found in singles sites was higher in the islands (66) than the mainland (35). The number of rare taxa found in single sites was also higher in the islands (62) than the mainland (24). Distribution, the number of taxa found in single sites and the number of rare taxa (RA < 5%) found in singles sites are presented in Figures 3 and 4.

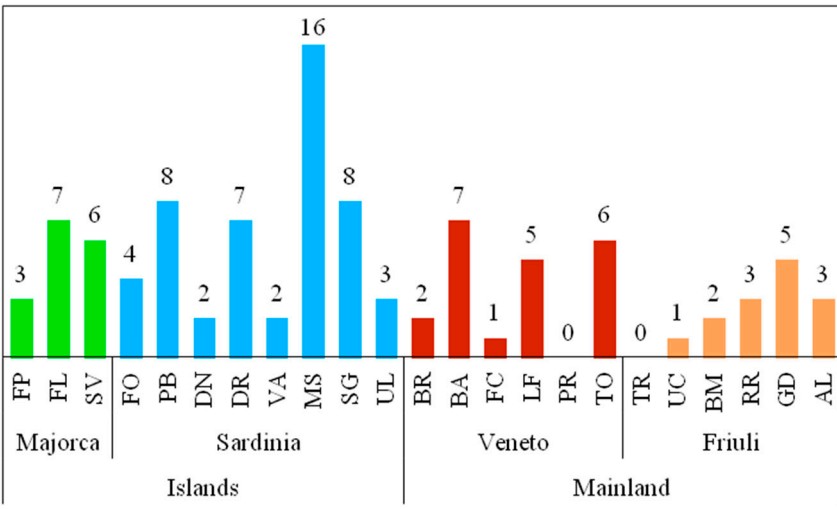

**Figure 3.** Distribution and number of taxa found in single sites. All codes of the sites are reported in Table 1.

**Table 3.** Floristic list and relative abundance of the most abundant diatom taxa (RA ≥ 5% in at least one site).

| Taxa | Majorca | | | | | | Sardinia | | | | | Veneto | | | | | | Friuli | | | | | |
|---|---|---|---|---|---|---|---|---|---|---|---|---|---|---|---|---|---|---|---|---|---|---|---|
| | FP | FL | SV | FO | PB | DN | DR | VA | MS | SG | UL | BR | BA | FC | LF | PR | TO | TR | UC | BM | RR | GD | AL |
| *Achnanthidium exile* (Kützing) Heiberg | 9.2 | | | | | | | | | | | | | | | | | | | | | | |
| *Achnanthidium lineare* W. Smith | | | | 1.7 | | | 0.9 | | | 0.6 | | | | | 0.8 | | | 7.7 | 2.4 | 0.5 | 0.5 | 5.4 | 7.0 |
| *Achnanthidium minutissimum* (Kützing) Czarnecki | 76.6 | 7.9 | 64.6 | 16.4 | 35.1 | 31.2 | 5.3 | 4.3 | 0.9 | 3.6 | 1.0 | 43.3 | 9.8 | 31.4 | 69.2 | 15.7 | | | | | | 4.4 | |
| *Achnanthidium pyrenaicum* (Hustedt) Kobayasi | 6.1 | | | | | | | | | | | 2.5 | | | 7.2 | | | 52.6 | 74.9 | 30.0 | 6.3 | 63.4 | 22.9 |
| *Amphora inariensis* Krammer | | | | | | | | | | | 0.6 | 7.8 | 12.3 | 40.2 | 1.6 | 1.6 | | | | | | | |
| *Amphora indistincta* Levkov | | | | 1.0 | | 0.5 | 3.0 | | 0.7 | 11.1 | 2.9 | | | | | | | | | | | | |
| *Amphora pediculus* (Kützing) Grunow | 2.1 | 27.0 | 1.2 | 6.9 | | 1.5 | 17.6 | 0.7 | 15.3 | 21.7 | 17.9 | 10.7 | 11.9 | 7.9 | 3.9 | | 0.2 | 1.0 | 0.5 | 0.5 | | | |
| *Caloneis fontinalis* (Grunow) Cleve-Euler | | | | 0.7 | | | 0.7 | | | 3.9 | 6.0 | 5.6 | | | 1.0 | | 0.8 | | | | | | |
| *Cocconeis euglypta* Ehrenberg | 0.9 | 14.3 | | 7.7 | 6.1 | | 7.5 | | 12.1 | 11.1 | 1.7 | | | | | | 0.2 | | | | | | |
| *Cocconeis pseudolineata* (Geitler) Lange-Bertalot | | | | | 1.6 | 46.2 | | | 0.5 | 7.2 | 0.5 | 9.4 | | | | | | | | | | 0.5 | 1.0 |
| *Cymbella affinis* Kützing | | | 1.4 | | | | | | | | | | | | | | | | | | 1.5 | 9.8 | |
| *Cymbella tridentina* Lange-Bertalot, M. Cantonati & A. Scalfi | | | | | | | | | | | | | | | | | | | | | | | 32.3 |
| *Derticula tenuis* Kützing | 0.2 | | | | | | | | | | | | | | 8.0 | | | 0.5 | 7.0 | | 0.5 | 2.0 | 1.5 |
| *Diploneis separanda* Lange-Bertalot | 0.2 | | 6.3 | | | | | | 0.5 | 2.1 | | | 0.4 | | | | | | | | | | |
| *Ellerbeckia arenaria* (Moore ex Ralfs) R.M. Crawford | | | | | | | 5.3 | | 0.7 | 1.5 | | | | | | | | | | | | | |
| *Encyonema minutum* (Hilse) D.G. Mann | | | | | | | | | | | | | 0.3 | | | | | | | 1.5 | 1.5 | 7.3 | |
| *Encyonema ventricosum* (Agardh) Grunow | | | | | | | | | | | | | | | | | | 0.5 | 11.1 | | 2.0 | 4.4 | |
| *Eunotia arcubus* Nörpel & Lange-Berlalot | | | 15.3 | | | | | | | | | | | | | | | | | | | | |
| *Fallacia insociabilis* (Krasske) D.G. Mann | | | | | | | | | 0.2 | | | | | | | | 20.0 | | | | | | |
| *Fallacia muraloides* (Hustedt) D.G. Mann | | | | | | | | | | | | | 5.7 | | | | | | | | | | |
| *Fallacia sublucidula* (Hustedt) D.G. Mann | | | | | | | | | 0.2 | 5.3 | | | | | | | | | | | | | |
| *Gomphonema* aff. *cymbelliclinum* Reichardt & Lange-Bertalot | | | | | | | | 30.2 | | | | | | | | | | | | | | | |
| *Gomphonema elegantissimum* Reichardt & Lange-Bertalot | | | | | | | | | | | | | | | | | | 26.8 | 5.8 | 47.5 | | | 3.0 |
| *Gomphonema micropus* Kützing | | | 0.5 | 6.2 | | | | 1.2 | | | 1.5 | | | | | 0.2 | | | | | | | |
| *Gomphonema rosenstockianum* Lange-Bertalot & Reichardt | | | 0.9 | | | 3.7 | | | | | 18.6 | | | | | | | | | | | | |
| *Gomphonema* sp.2 | | | | | | | | | | | | | | | | | | | | 12.0 | | | |
| *Gomphonema subclavatum* Grunow | | | | | | | 0.5 | | | | | | | | | 0.2 | | | | | | 1.5 | 17.4 |
| *Gomphonema tergestinum* (Grunow) Fricke | | | | | | | | | | | | | | | | | | 7.2 | 1.9 | | 6.3 | | |
| *Gomphosphenia lingulatiformis* (Lange-Bertalot & Reichardt) Lange-Bertalot | | | | | | | | | | | | | 32.7 | 0.6 | | | | | | | | | |
| *Hannaea arcus* (Ehrenberg) R.M. Patrick | | | | | | | | | | | | | | | | | | | | | | 10.2 | |
| *Humidophila contenta* (Grunow) Lowe, Kociolek et al. | | 3.7 | | 0.2 | | | 0.2 | | 5.9 | | 3.6 | | 2.0 | | | 1.6 | 57.3 | | | | | | |
| *Kolbesia gessneri* (Hustedt) Aboal | | | | | | | 9.6 | | 12.3 | 1.9 | | | | | | | | | | | | | |
| *Meridion circulare* (Greville) C. Agardh | | | 0.2 | 5.7 | | 1.7 | 0.2 | 61.2 | | | 16.2 | | | | | 0.6 | | | | | | 41.5 | 3.0 |
| *Nitzschia frustulum* (Kützing) Grunow | | | | | | | | | | | | | | | | | 7.4 | | | | | | |
| *Nitzschia inconspicua* Grunow | | 0.7 | | 17.4 | 0.9 | 0.2 | 0.5 | | 2.1 | | | | | | | | | | | | | | |
| *Nitzschia recta* Hantzsch ex Rabenhorst | | | | | | | 0.2 | | 5.2 | | | | | | | | | | | | | | |
| *Odontidium mesodon* (Ehrenberg) Kützing | | | | | | | | | | | | | | | | | 0.2 | 0.5 | 0.5 | | 18.0 | 0.5 | 0.5 |
| *Planothidium frequentissimum* (Lange-Bertalot) Lange-Bertalot | | 9.7 | 3.2 | 4.5 | 28.6 | 1.2 | 2.5 | | 1.8 | | 16.0 | 1.3 | 0.7 | 1.3 | 0.6 | 8.0 | 2.3 | | | | | | |
| *Planothidium lanceolatum* (Brébisson ex Kützing) Lange-Bertalot | | | 12.9 | | | | 0.7 | | 0.9 | 6.0 | 13.6 | 4.4 | 0.4 | 1.5 | 0.0 | 1.9 | | | | | | 0.5 | 0.5 |
| *Platessa conspicua* (A. Mayer) Lange-Bertalot | | 11.9 | | | | | | | | | | | | | | | | | | | | | |
| *Psammothidium grischunum* (Wuthrich) Bukhtiyarova & Round | | | | | | | | | | | | | | | 7.7 | | | | | | | | |
| *Pseudostaurosira alvareziae* Cejudo-Figueras, E.A. Morales & Ector | | | | | | | 12.3 | | 5.7 | | | | | | | | | | | | | | |
| *Rhoicosphenia abbreviata* (C. Agardh) Lange-Bertalot | | 1.8 | | | | 0.2 | | | 5.9 | | 0.9 | | | | | | | | | | | | |
| *Sellaphora atomoides* (Grunow) C.E. Wetzel & Van de Vijver | | | 1.7 | | | | | | | | | | | | 1.4 | 20.8 | | | | | | | |
| *Sellaphora nigri* (De Notaris) C.E. Wetzel & Ector | 0.2 | 13.2 | 10.2 | 0.7 | | | 5.5 | | | | | 11.6 | 0.9 | 9.0 | 0.0 | 26.3 | | 1.0 | | | | | |
| *Sellaphora seminulum* (Grunow) D.G. Mann | | | | | | | 2.3 | | | | | 1.3 | 8.2 | 0.6 | 0.0 | 21.7 | | | | | | | |

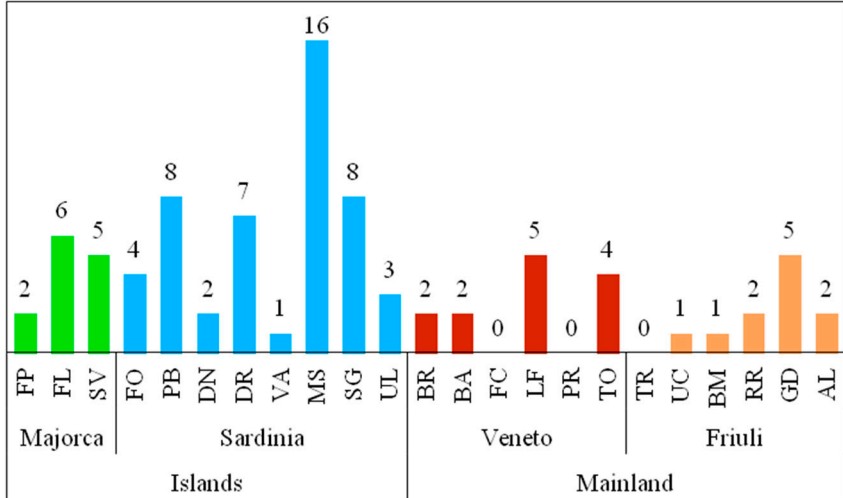

**Figure 4.** Distribution and number of rare taxa (RA < 5%) found in single sites. All codes of the sites are reported in Table 1.

The results of the structural indices are reported in Table 4. Species richness varied from 8 at VA to 54 at MS, both in Sardinia. The minimum and maximum values of the Shannon-Wiener diversity index ($H'$) were recorded at FP (1.40) in Majorca and at MS (4.48) in Sardinia. The values of Pielou's evenness index ($J'$) ranged from 0.31 at UC, in Friuli to 0.81 at DR, in Sardinia and BA, in Veneto.

**Table 4.** Values of species richness (R), Shannon-Wiener diversity index (H') and Pielou's evenness index (J') for diatom assemblages of the springs studied.

| Spring | R | H′ | J′ |
| --- | --- | --- | --- |
| **Majorca** | | | |
| FP | 14 | 1.40 | 0.37 |
| FL | 19 | 3.19 | 0.75 |
| SV | 16 | 1.94 | 0.49 |
| **Sardinia** | | | |
| FO | 27 | 3.77 | 0.79 |
| PB | 22 | 2.93 | 0.66 |
| DN | 18 | 2.21 | 0.53 |
| DR | 43 | 4.37 | 0.81 |
| VA | 8 | 1.44 | 0.48 |
| MS | 54 | 4.48 | 0.78 |
| SG | 37 | 4.05 | 0.78 |
| UL | 25 | 3.22 | 0.69 |
| **Veneto** | | | |
| BR | 16 | 2.51 | 0.76 |
| BA | 21 | 2.90 | 0.81 |
| FC | 10 | 2.00 | 0.77 |
| LF | 20 | 1.49 | 0.45 |
| PR | 13 | 2.47 | 0.78 |
| TO | 16 | 1.83 | 0.58 |
| **Friuli** | | | |
| TR | 20 | 1.94 | 0.47 |
| UC | 26 | 1.43 | 0.31 |
| BM | 35 | 1.91 | 0.39 |
| RR | 19 | 2.80 | 0.67 |
| GD | 34 | 2.09 | 0.43 |
| AL | 22 | 2.76 | 0.62 |

### 3.3. Comparison among Diatom Assemblages

The nMDS ordination showed a clear separation of the diatom assemblages in the springs of Friuli from those of all other regions (Figure 5). The springs RR in Friuli, VA in Sardinia and TO in Veneto were also quite separated from their respective groups. Significant differences among assemblages were confirmed by the ANOSIM test (global R = 0.610; *p* = 0.001) and pairwise tests (Table 5).

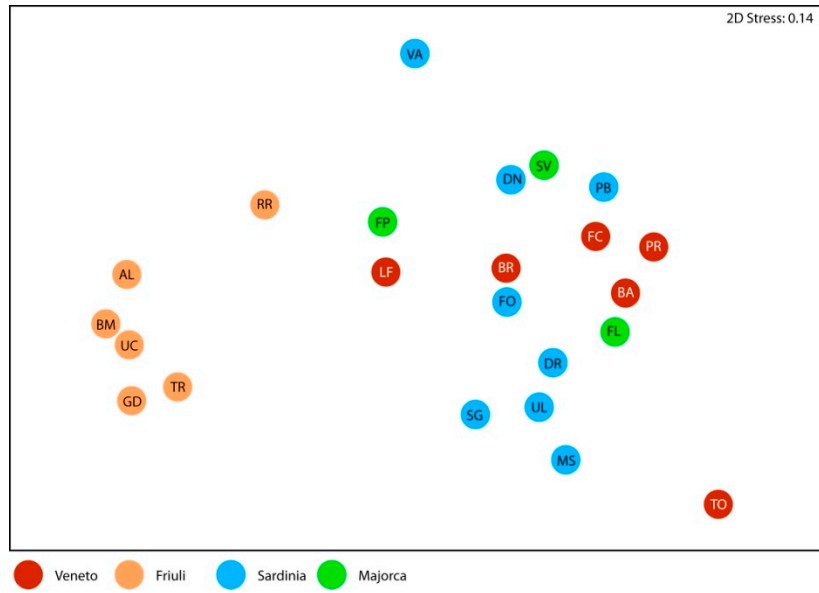

**Figure 5.** Non-metric multidimensional scaling (nMDS) ordination plot for diatom assemblages from the springs studied. All codes of the sites are reported in Table 1.

**Table 5.** Results of a one-way analysis of similarities (ANOSIM) test evaluating global and pairwise differences of the diatom assemblages in the four geographic regions. Significant values are reported in bold.

| Groups | R Statistic | *p* Significance Level |
|:---:|:---:|:---:|
| Global effect | **0.610** | **0.001** |
| Pairwise tests: | | |
| **Veneto vs Sardinia** | **0.278** | **0.008** |
| Veneto vs Majorca | 0.068 | 0.333 |
| **Veneto vs Friuli** | **0.898** | **0.002** |
| Sardinia vs Majorca | 0.052 | 0.358 |
| **Sardinia vs Friuli** | **0.928** | **0.001** |
| **Majorca vs Friuli** | **0.963** | **0.012** |

The *t*-test, performed on the structural indices, indicated significant differences only in diversity as expressed by Shannon-Wiener's index (*t* = 2.304; *p* = 0.031).

### 3.4. Relationships with Environmental Variables

In the CCA analysis (Figure 6), the first two axes accounted for 60.5% of the total variance of diatom species and environmental data (axis 1: 44.8% and axis 2: 15.7%). Ordination data distinguished three main groups of sites and species. The first group, situated in the right side of the plot includes the sites of the Friuli, with higher discharge and more alkaline waters. This group is composed of *Achnanthidium lineare*, *A. pyrenaicum*, *Denticula tenuis*, *Gomphonema elegantissimum*, and *G. tergestinum*. The second group, situated at the upper left part of the plot contains the sites of the islands, characterized by higher temperature and mineralization of water (Cl⁻ and conductivity). This group is composed of *Amphora pediculus*, *Caloneis fontinalis*, *Cocconeis euglypta*, *C. pseudolineata*, *Meridion circulare*, *Humidophila contenta*,

*Kolbesia gessneri, Planothidium frequentissimum,* and *Pseudostaurosira alvareziae.* The third group, situated at the bottom left part of the plot, include the sites of the Veneto, characterized by higher shading and impacts. This group is composed of *Amphora inariensis, Sellaphora nigri,* and *S. seminulum.* Overall, the significant variables for diatom assemblages were discharge ($p = 0.018$) and Cl$^-$ ($p = 0.002$).

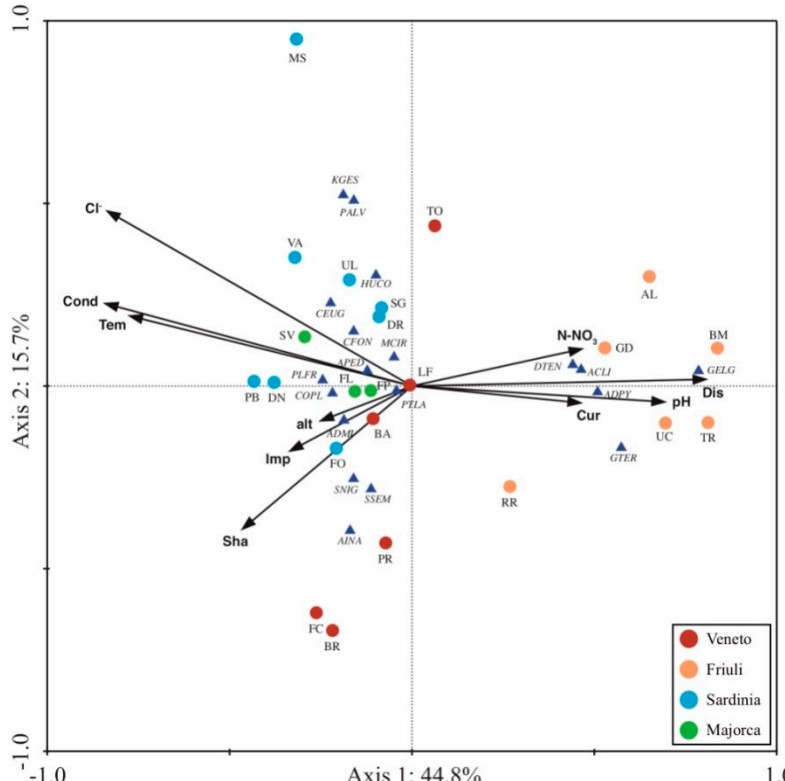

**Figure 6.** Canonical Correspondence Analysis (CCA) ordination plot for the diatom assemblages. Vectors = environmental variables; circles = spring sites, triangles = diatom species. All codes of sites are reported in Table 1. Acronyms of the species: ACLI = *Achnanthidium lineare*; ADMI = *Achnanthidium minutissimum*; ADPY = *Achnanthidium pyrenaicum*; AINA = *Amphora inariensis*; APED = *Amphora pediculus*; CFON = *Caloneis fontinalis*; CEUG = *Cocconeis euglypta*; COPL = *Cocconeis pseudolineata*; DTEN = *Denticula tenuis*; GELG = *Gomphonema elegantissimum*; GTER = *Gomphonema tergestinum*; HUCO = *Humidophila contenta*; KGES = *Kolbesia gessneri*; MCIR = *Meridion circulare*; PLFR = *Planothidium frequentissimum*; PTLA = *Planothidium lanceolatum*; PALV = *Pseudostaurosira alvareziae*; SNIG = *Sellaphora nigri*; SSEM = *Sellaphora seminulum*.

The IndVAL analysis identified a total of eight indicator species for the three distinct groups of sites based on the CCA analysis (Figure 7, Table 6).

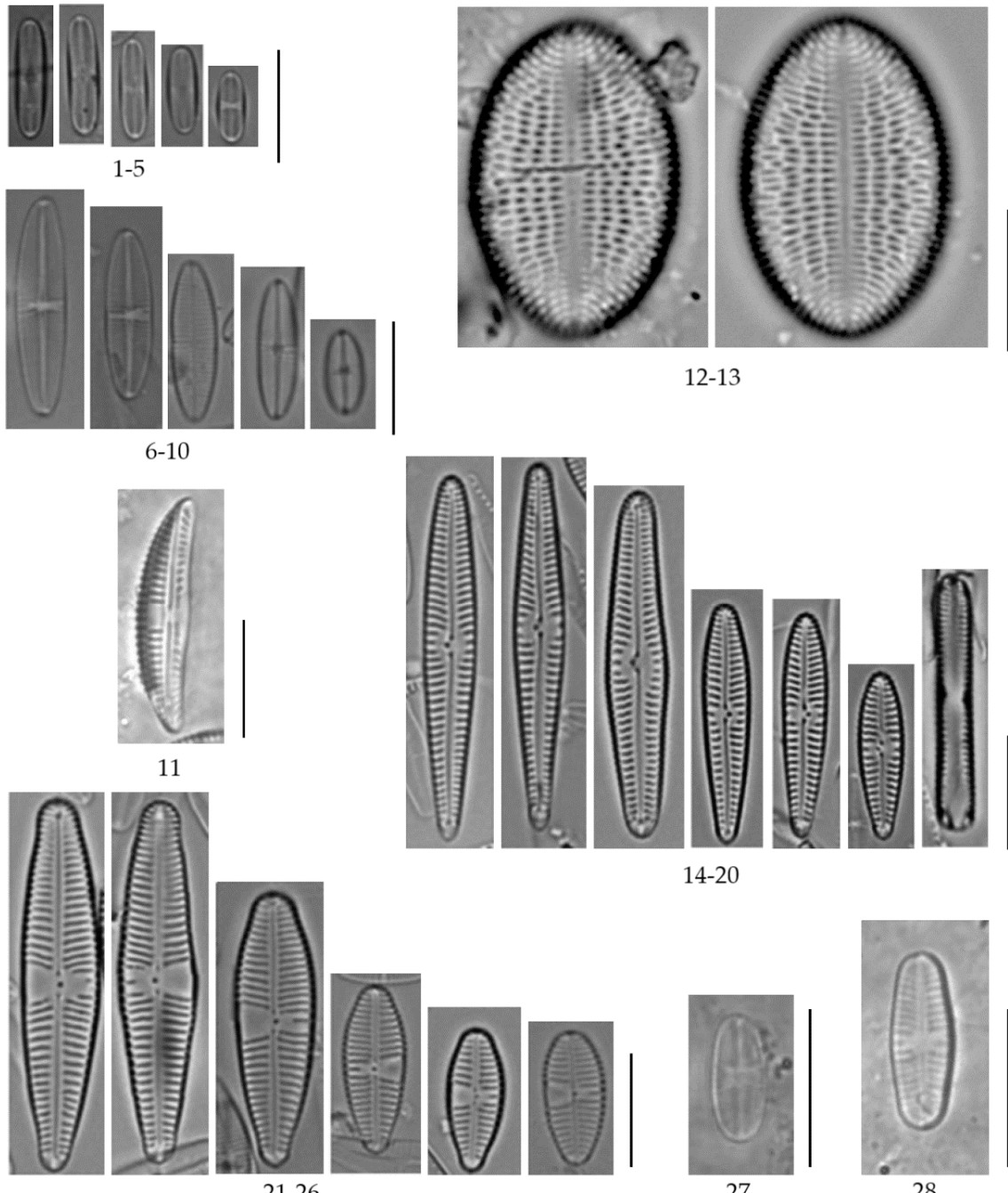

**Figure 7. 1–28.** Light microscopy (LM). 1–5: *Achnanthidium lineare* W. Smith; 6–10: *Achnanthidium pyrenaicum* (Hustedt) Kobayasi; 11: *Amphora inariensis* Krammer; 12–13: *Cocconeis euglypta* Ehrenberg; 14–20: *Gomphonema elegantissimum* Reichardt & Lange-Bertalot; 21–26: *Gomphonema tergestinum* (Grunow) Fricke; 27: *Sellaphora nigri* (De Notaris) C.E. Wetzel & Ector; 28: *Sellaphora seminulum* (Grunow) D.G. Mann. Scale bars = 10 μm.

**Table 6.** Indicator taxa from the three groups of sites visualized after the CCA according to indicator species analysis (IndVal).

| Groups of Sites | Species | Code | IndVal | *p*-Value |
|---|---|---|---|---|
| **1 (Friuli)** | *Achnanthidium pyrenaicum* | ADPY | 0.975 | 0.001 |
| | *Achnanthidium lineare* | ACLI | 0.950 | 0.002 |
| | *Gomphonema elegantissimum* | GELG | 0.816 | 0.005 |
| | *Gomphonema tergestinum* | GTER | 0.707 | 0.018 |
| **2 (Majorca + Sardinia)** | *Cocconeis euglypta* | CEUG | 0.850 | 0.003 |
| **3 (Veneto)** | *Amphora inariensis* | AINA | 0.911 | 0.001 |
| | *Sellaphora seminulum* | SSEM | 0.801 | 0.007 |
| | *Sellaphora nigri* | SNIG | 0.791 | 0.027 |

## 4. Discussion

### 4.1. Main Characteristics of the Diatom Assemblages

Overall, the taxa abundant and common to more than six sites were *Achnanthidium lineare*, *A. minutissimum*, *A. pyrenaicum*, *Amphora pediculus*, *Caloneis fontinalis*, *Cocconeis euglypta*, *C. pseudolineata*, *Denticula tenuis*, *Humidophila contenta*, *Meridion circulare*, *Planothidium frequentissimum*, *P. lanceolatum*, and *Sellaphora nigri*. Most of these taxa were found in karst springs of different geographic areas in Europe, such as France [68], Poland [69], Austria [70], Slovenia [21], and Bosnia and Herzegovina [23]. By contrast, a large number of taxa occurred in single sites and most of them were rare (RA < 5%). Our observations are supported by previous studies on springs in Italy [32,36] and France [68]. The high number of taxa found in single sites suggests a general marked heterogeneity of assemblages but also a specificity of each site at a regional level, since these species often have high affinity with specific abiotic conditions [32]. This aspect also underlines the role of the springs as refuges for many different species as recently reported by Taxböck et al. [7]. Several species with a strong affinity to spring environments or crenophilous species were found: *Caloneis fontinalis*, *Cymbella tridentina*, *Diploneis krammeri* Lange-Bertalot & Reichardt, *Eunotia arcubus*, *Eunotia minor* (Kützing) Grunow, *Geissleria gereckei* Cantonati & Lange-Bertalot, *Gomphonema elegantissimum*, *Gomphonema lateripunctatum* Reichardt & Lange-Bertalot, *Meridion circulare*, and *Odontium mesodon*. Most of them were recorded in karst springs both in high and low-altitude, with different disturbance levels e.g., [18,71,72]. Further, *Meridion circulare* and *Odontium mesodon* constitute a common association of species of karst springs of different geographic areas. In our study, the crenophilous species were generally more abundant in springs with low impact: *C. fontinalis* at SG and BR (score = 3–5), *C. tridentina* at AL (score = 2), *E. arcubus* at SV (score = 3) *G. elegantissimum* at BM (score = 2), *M. circulare* at UL, VA and RR (score = 3) and *O. mesodon* at RR (score = 3). *Cymbella tridentina*, found only at AL (1249 m a.s.l.) in Friuli, seems to be the only crenophilous species with a distribution restricted to high-altitude springs in good accordance with previous studies that reported this species from the uppermost sections of carbonate spring-fed streams of the Alps at elevations >1200 m a.s.l. [18,73]. Many taxa are not closely linked to the aquatic environment according to Van Dam et al. [59]. Examples of the pseudaerial/euaerial species, belonging to categories 4+5, (i.e., species occurring in temporarily dry places or outside water bodies) were *Achnanthes coarctata* (Brébisson ex W. Smith) Grunow, Cosmioneis pusilla (W. Smith) D.G. Mann & Stickle, *Denticula subtilis* Grunow, *Humidophila gallica* (W. Smith) Lowe, Kociolek, Q. You, Q. Wang & Stepanek, *Diploneis oblongella* (Nägeli ex Kützing) Cleve-Euler, *Ellerbeckia arenaria*, *Fallacia insociabilis*, *Eunotia arcubus*, *Humidophila contenta* and *Nitzschia valdestriata* Aleem & Hustedt. Whilst being absent in the springs of Friuli, characterized by high discharge, these species were observed

mainly in springs with low discharge in Sardinia and Veneto. In these sites, these species can find suitable parts of substrata, temporarily uncovered by the water film due to the low discharge and water abstraction. Their presence underlines the role of springs as multiple ecotones, in particular, the land-water transition.

The diatom flora of the springs studied was composed by a small group of centric species rarely observed in spring environments: *Aulacoseira* sp., *Cyclotella meneghiniana* Kützing, *Ellerbeckia arenaria*, *Melosira varians* C. Agardh, and *Pleurosira laevis* (Ehrenberg) Compère. These centric diatoms were found only in the Sardinian springs, generally with very low abundance, and could be favoured by low current velocity. Similar findings were reported in previous studies on thermo-mineral springs of the island [74,75].

### 4.2. Structure of the Diatom Assemblages

In general, the springs studied hosted rich and diversified diatom communities with a balanced distribution of the species as expressed by species richness (*R*), diversity (*H'*), and evenness (*J'*) indices. High species richness (≥25 taxa) was found at several sites, in Sardinia and Friuli, both with higher naturalness and strongly modified by water abstraction systems. In fact, the coexistence of numerous species seems to depend on a complex combination of local abiotic factors and contrastingly disturbed microhabitats, besides the degree of naturalness of the site [8,32]. The values of species richness were comparable among spring types, i.e., R-L vs LR, except for the spring SG, which showed a higher number of species. This is in agreement with an extensive study carried out on different spring types in the southeastern Alps [76].

The *t*-test performed on structural indices revealed significant differences between islands and the mainland only in diversity as expressed by the Shannon index (*H'*).

### 4.3. Comparison of Diatom Assemblages from Islands and Mainland

The nMDS ordination and ANOSIM test performed on the most abundant taxa did not show a real distinctiveness among diatom assemblages from islands and the mainland. However, the diatom assemblages from the sites of Friuli formed a well-separated cluster and revealed high dissimilarity with those from all other regions. Differences in this group of springs seem to be attributable to specific local factors, such as hydrological features (high discharge and current velocity) and lower mineralization level of the water. In fact, these factors are considered among the most important drivers for the growth and distribution of diatom species [17]. In the nMDS plot, the springs RR, VA and TO were also quite separated from their groups, respectively in Friuli, Sardinia, and Veneto. In Sardinia, VA was the only spring with a seasonal regime. The water flow permanence was an important driver for diatom flora in this site as reported in a previous study [32]. Further, this factor was indicated as a relevant local hydrological factor for diatom assemblages in springs [8,77]. By contrast, differences for the springs RR and TO seem less clear and are probably due to a combination of different factors.

Considering the whole data set, we found further differences in diatom assemblages from the islands and the mainland. For example, the number of taxa found in single sites and the number of rare taxa (also found in single sites) were higher in springs of the islands than those of the mainland. This suggests that the role of springs as refuges for rare species may be greater in the islands and that the condition of geographic insularity may be important in maintaining a high level of biodiversity. In fact, a limited number of sites and low abundance were indicated as characteristics of rare diatom species [78]. In addition, springs are considered in themselves water islands with specific and well-differentiated biocoenoses due to their scattered distribution [8–19] and the presence of species with restricted distributions are expected in more isolated freshwater systems because of geographic, physical, and chemical barriers [8].

*4.4. Relationships Diatoms-Environmental Variables*

Among the environmental variables measured and analysed in this study, discharge and Cl⁻, respectively associated with the sites of the islands and Friuli, explained the significant amount of variance in diatom assemblages according to CCA analysis. Our results seem to reflect well the characteristics of these sites. For example, water mineralization was reported as an important feature of the springs in Majorca Island [26,79]. In addition, the low discharge was indicated among the most important environmental variables determining the diatom assemblages in low-elevation carbonate springs [72].

The IndVal analysis revealed indicator species quite consistent with the two major gradients highlighted by the CCA analysis. The indicator taxa for the springs of Friuli (group 1) were *Achnanthidium pyrenaicum, A. lineare, Gomphonema elegantissimum* and *G. tergestinum.* The first two species of *Achnanthidium* genus were observed in all sites of Friuli. These species are reported as well-adapted at high discharge and current velocity in the literature e.g., [45,80]. *Gomphonema tergestinum* is also reported as a species present in fast-flowing running waters [45] while *G. elegantissimum* seems to prefer moderate current velocity [80] but in these springs may be favoured by the availability of nutrients. For the springs of the islands (group 2) *Cocconeis euglypta* was the only indicator species. This species seems to be typical of flowing waters with medium-high mineralization [81,82]. However, habitat and ecology of this species are not yet well known because it was not consistently identified over time e.g., [45]. The springs of Veneto (group 3) were characterized by *Amphora inariensis, Sellaphora seminulum* and *S. nigri*. *Amphora inariensis* was found in undisturbed and sometimes moderately impacted freshwater environments [45]. *Sellaphora seminulum* was reported in springs and intermittently-drying small water bodies but the ecology of this species is not still precisely known [45]. Instead, *S. nigri* seems to be related to conditions of greater anthropogenic impact and was indicated as a well-known indicator of nutrient enrichment [51]. This species was reported in low-elevation carbonate springs affected by anthropogenic disturbance [72]. However, NGS data suggest great genetic diversity, and the consequent likely necessity of future revisions of its taxonomy and ecology [45].

## 5. Conclusions

This study provides the first comparison of diatom biodiversity in karst springs with different anthropogenic impact levels in two Mediterranean geographic areas with contrasting characteristics (islands and mainland). In general, the biotic integrity level was good both in sites with greater naturalness and in those strongly modified by water abstraction systems. However, crenophilous species were generally present and abundant in less impacted springs. The diatom flora was very heterogeneous and significant differences were found in the diversity of assemblages between islands and the mainland. The springs in the islands also recorded the highest number of species observed in single sites, as well as of rare species in single sites. Based on the most abundant species, significant differences were observed according to our initial hypothesis. Discharge and Cl⁻ were two important variables in the springs studied and significant factors influencing diatom assemblages. Our results confirm the role of springs as refuges for rare species and suggest that the geographic insularity may be an important factor in maintaining diatom biodiversity. However, Mediterranean karst springs are fragile ecosystems increasingly exposed to natural and anthropogenic pressures. Morphological alterations and uncontrolled water abstraction as a result of inadequate management may cause a reduction of their ecological integrity and a loss of biodiversity in the future.

**Supplementary Materials:** The supplementary materials are available online at http://www.mdpi.com/2073-4441/11/12/2602/s1.

**Author Contributions:** G.G.L. and M.C. conceived and designed the paper. G.G.L., S.B., C.D., and M.C. carried out sampling and identification of the diatom species. R.Z. and E.P. provided environmental data. B.M.P. performed all statistical analyses and figures. G.G.L. wrote the paper and prepared tables and figures. G.G.L., B.M.P., C.D., A.L. and M.C. revised and edited the manuscript.

**Funding:** This research received no external funding. The activities of Prof. Antonella Lugliè were supported by the research fund of the University of Sassari (fondo di Ateneo per la ricerca 2019).

**Acknowledgments:** We thank Giuseppe Cossu, Salvatore Sulis (Sardinia Forestry Corps) and EN.A.S. (Ente Acque della Sardegna) for the field inspections, Bruno Giordanu and Maria Antonietta Mariani for their support in sampling and the colleagues of the University of Sassari (Laboratory of Aquatic Ecology) for the analysis of water samples. We are also grateful to the colleagues of ARPA FVG (Regional Agency for Environmental Protection of Friuli-Venezia Giulia), in particular Ivan Martinuzzi for the analysis of water samples.

**Conflicts of Interest:** The authors declare no conflict of interest.

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
