# Peer review of "Diatom Biodiversity in Karst Springs of Mediterranean Geographic Areas with Contrasting Characteristics: Islands vs Mainland"

_water, doi:10.3390/w11122602_

Round 1

Reviewer 1 Report

Overall I do not see any major issue with the paper but I have some minor concerns/remarks:

- The idea of eliminating the rare species from the analysis to reduce the noise is understandable but in my opinion here, given the tools used, unnecessary (CCA in particular is quite robust to that kind of minor noise). In any cases, the table of relative abundances including the rare species eliminated from the analysis should be provided as supplementary material (will be useful for further research: although rare species were not crucial for this particular study, they are to our understanding of biodiversity patterns in general).
- Many references to the statistical tools used are missing, specifically those for: paired t-test [Student, 1908; Fischer, 1925], Shapiro-Wilk test [Shapiro and Wilk, 1965], CCA [ter Braak, 1986], DCA [Hill and Gauch, 1980], NMDS [Kruskal, 1964], Bray-Curtis dissimilarity [Bray and Curtis, 1957].
- I think the species identified as indicator species should be presented in a plate, so that we know what the authors considered those species to be.

Some additional line-by-line comments:
l. 22: "with different impact levels" should specify you're referring to anthropogenic impact, it's not clear as it stands in the abstract.
l.35: "the most relevant " for what?
Table 1: Longitude (W) : ... isn't it (E) though? (as in East of Meridian 0).
l.160: "For this analysis, p <0.03 was considered significant." I this a typo? If not, this is very unusual, please explain.
l. 168: "(499 permutations)." Why? Why not 500 or 1000?
l. 275 "5. Discussion" misnumbered, should be 4
l. 318: "The springs studied showed a good level of biotic integrity as expressed by species richness (R), diversity (H'), and evenness (J') indices." Not sure what the authors mean by that

Student (1908). "The probable error of a mean". Biometrika, 6 (1): 1-25.
Fisher R. A. (1925) "Applications of 'Student's' distribution" Metron (5): 90-104.
Bray, J. R. and J. T. Curtis. (1957). An ordination of upland forest communities of southern Wisconsin. Ecological Monographs 27:325-349.
Kruskal, J.B. (1964). Nonmetric multidimensional scaling: a numerical method. Psychometrika 29, 115–129.
Shapiro, S. S. and Wilk, M. B. (1965). "An analysis of variance test for normality (complete samples)". Biometrika. 52 (3–4): 591–611.
Hill, M.O. and Gauch, H.G. (1980). Detrended Correspondence Analysis: An Improved Ordination Technique. Vegetatio 42, 47–58.
Ter Braak, C. J. (1986). Canonical correspondence analysis: a new eigenvector technique for multivariate direct gradient analysis. Ecology, 67(5):1167–1179.

Author Response

[Water] Manuscript ID: water-642948 - Minor Revisions

Review 1.

Comments and Suggestions for Authors

Overall I do not see any major issue with the paper but I have some minor concerns/remarks:

- The idea of eliminating the rare species from the analysis to reduce the noise is understandable but in my opinion here, given the tools used, unnecessary (CCA in particular is quite robust to that kind of minor noise). In any cases, the table of relative abundances including the rare species eliminated from the analysis should be provided as supplementary material (will be useful for further research: although rare species were not crucial for this particular study, they are to our understanding of biodiversity patterns in general).

In this study the rare species have been very numerous and often present in only one site. For this reason we have preferred to exclude them a priori from the analysis. We have provided the complete table of species (see supplementary material) as suggested. We agree that this can be useful for understanding of biodiversity patterns.

- Many references to the statistical tools used are missing, specifically those for: paired t-test [Student, 1908; Fischer, 1925], Shapiro-Wilk test [Shapiro and Wilk, 1965], CCA [ter Braak, 1986], DCA [Hill and Gauch, 1980], NMDS [Kruskal, 1964], Bray-Curtis dissimilarity [Bray and Curtis, 1957].

Thank you for this observation and for kindly providing us these references. We added them in the text (see lines 157, 167, 168, 173, 174) and in References section (see lines 592, 593, 594, 600, 601, 607, 609).

- I think the species identified as indicator species should be presented in a plate, so that we know what the authors considered those species to be.

We agree. We have done it (see new Figures 7-34).

Some additional line-by-line comments:

22: "with different impact levels" should specify you're referring to anthropogenic impact, it's not clear as it stands in the abstract.

We have specified this (see line 22).

l.35: "the most relevant " for what?

We have re-phrased this part (see lines 36-38).

Table 1: Longitude (W): isn't it (E) though? (as in East of Meridian 0).

We have corrected this (see Table 1).

l.160: "For this analysis, p <0.03 was considered significant." I this a typo? If not, this is very unusual, please explain.

We have arbitrarily reduced the commonly used significance level in the text. We have corrected to 0.05 (= 5%).

168: "(499 permutations)." Why? Why not 500 or 1000?

We used the default software settings.

275 "5. Discussion" misnumbered, should be 4

We have corrected this (see lines 308, 309, 352, 364, 388).

318: "The springs studied showed a good level of biotic integrity as expressed by species richness (R), diversity (H'), and evenness (J') indices." Not sure what the authors mean by that

We have better explained this concept (see lines 353-355).

Other changes in the manuscript are reported in red color.

Reviewer 2 Report

The manuscript „Diatom biodiversity in karst springs of Mediterranean geographic areas with contrasting characteristics: islands vs mainland“ is well written and covers an important aquatic habitat.

Overall I can suggest this manuscript for publication, but I recommend MINOR REVISIONS in order to improve style and quality. I think there are some points that should be included, thus please consider the following suggestions:

Line 109: how much water is captured at RR? Is there a minimum flow / eflow to ensure a functional ecosystem? Chapter 2.2: where did you samples? Directly at the source, max. XXm from the source, … maybe you could include typical photographs for selected sampling points Table 1: could you add information on mean annual flow? Table 2: this is the discharge (class) during the sampling? Line 133: delete “Angeli et al., Delgado et al., and Lai et al.“ à format = [xx] à also applies e.g. in line 121, 122, … Chapter 5.1: Start with „overall…“ the first 2 sentences (277-282) can be placed later – in a suitable position, e.g. at the end of this paragraph… Reference 20 & 21 is identical

Author Response

[Water] Manuscript ID: water-642948 - Minor Revisions

Review 2.

Line 109: how much water is captured at RR? Is there a minimum flow/eflow to ensure a functional ecosystem?

Several springs in this study are captured for drinking purposes, not just RR. We do not know the amount of water captured in these springs. These informations are certainly available at local managing bodies but not in literature. The definition of a minimum flow/eflow to ensure a functional ecosystem requires the use of specific hydromorphological and biological approaches for a period of at least one year. However, there are no extensive studies of these springs over time. Moreover, several of these springs were captured for a long time long ago, without considering this aspect.

Chapter 2.2: where did you samples? Directly at the source, max. XXm from the source, … maybe you could include typical photographs for selected sampling points.

We have better specified this in the text (see lines 130-131). We have also added some typical photographs for selected sampling points as suggested (see line 133 and new Figure 2).

Table 1: could you add information on mean annual flow?

Unfortunately these informations are not available for the springs studied (see above).

Table 2: this is the discharge (class) during the sampling?

Yes. The discharge (classes) reported in table 2 refer to the sampling period. We have specified this also in the text (see line 127).

Line 133: delete “Angeli et al., Delgado et al., and Lai et al.“ à format = [xx] à also applies e.g. in line 121, 122, …

We have done it (see lines 121, 123, 124, 136).

Chapter 5.1: Start with "overall…" the first 2 sentences (277-282) can be placed later – in a suitable position, e.g. at the end of this paragraph…

We have done it (see lines 346-351).

Reference 20 & 21 is identical

We have corrected this (see lines 496-499).

Other changes in the manuscript are reported in red color.
